# Market Women’s Perspectives on Coronavirus Disease 2019 (COVID-19): The Case of Ghana and South Africa

**DOI:** 10.3390/ijerph19159658

**Published:** 2022-08-05

**Authors:** Ebenezer Kwabena Frimpong, Peter Yamoah, Ebenezer Wiafe, Patrick Hulisani Demana, Moliehi Matlala

**Affiliations:** 1Indigenous Knowledge Systems Centre, Faculty of Natural and Agricultural Sciences, North-West University, Private Bag X 2046, Mmabatho 2790, South Africa; 2School of Pharmacy, The University of Health and Allied Sciences, Ho PMB 31, Ghana; 3Pharmacy Department, Ho Teaching Hospital, Ho MA 374, Ghana; 4Sefako Makgatho Health Sciences University, Molotlegi St, Ga-Rankuwa Zone 1, Ga-Rankuwa 0208, Medunsa 0204, South Africa

**Keywords:** COVID-19 pandemic, market women, negative economic impact, loss of revenue, Africa, gender perspective

## Abstract

Globally, countries are still battling health challenges and the negative economic stress on the citizenry caused by COVID-19. This study explored the perspectives of market women in Ghana and South Africa on COVID-19. Data collection was executed in both Ghana and South Africa between March 2021 and December 2021. Employing semi-structured questionnaires, face-to-face interviews were conducted. Most of the market women in Ghana described COVID-19 as a global pandemic, while market women in South Africa described the disease as the deadly flu. There were similarities in the perceived signs and symptoms of COVID-19. Market women in both countries specifically observed that not adhering to the safety protocols was the major mode of transmission. Lemon, garlic and ginger were the most common foodstuffs used by the market women to fight COVID-19. To prevent COVID-19 at their places of work, market women stressed the importance of observing the safety protocols. An overwhelming majority of market women in both countries bemoaned the negative impact of COVID-19 on their businesses and suggested the need for financial assistance from their respective governments. The findings are intended to assist policymakers in both Ghana and South Africa to implement interventional projects to assist women whom the literature suggests are the most vulnerable during pandemics such as COVID-19.

## 1. Introduction

The novel coronavirus disease was named severe acute respiratory syndrome coronavirus 2 (SARS-CoV-2) on 11 February 2020, by experts of the international committee on the taxonomy of viruses. SARS is a zoonotic coronavirus and falls under the β-coronavirus cluster [1]. The signs and symptoms of SARS-CoV-2 disease also known as coronavirus disease 2019 (COVID-19) are fever, cough diarrhoea and vomiting [1] and the major complications are acute respiratory, heart, anaemia and secondary infections [2]. From March 2020 when the WHO declared COVID-19 as a pandemic till February 2022, Ghana had recorded 157,751 active cases and 1419 deaths while South Africa had documented 3,634,811 active cases and 96,705 deaths [3]. COVID-19 has contributed to economic recession on the African Continent leading to a decline in foreign exchange receipts, unemployment and high food prices, etc. [4]. Market women play a significant role in the socio-economic development of the respective countries on the African continent [5,6]. In Ghana, a study found that the contributions of market women to the socio-economic development in the country are highly significant. About 30 women who sell mainly foodstuffs, clothes and plastic wares in Dunkwa and Twifo Praso markets in the central region of Ghana were interviewed in this study. The study revealed that monies generated from the market in the form of taxes serve as the main source of internally generated income that supports the local government developmental projects in both Dunkwa and Twifo Praso District Assemblies [5]. Similarly, in South Africa, most of the market women sell consumer goods (traditional medicine, clothes and foodstuff) by the wayside. Considering their contributions (generation of revenue) to the local economy in Durban, Ethekwini Municipality, South Africa has integrated them into the planning of the city by the provision of space and tables for the display of their goods for sale [6]. COVID-19 is expected to impact this group more and is unfortunately expected to curtail the gains made on gender equality and gender empowerment in the recent past in southern Africa [7]. This can be attributed to the fact that the informal labour force of Africa lacks insurance for any lost income [8].

In our quest to fight COVID-19 in our societies, we cannot ignore the contributions of market women. They are considered essential workers who ensure that foodstuffs and other commodities are readily available for the general population amid the COVID-19 pandemic. A study conducted to determine the community drivers affecting adherence to WHO guidelines against COVID-19 amongst rural traders in Uganda revealed that most of the participants (predominantly females) had sufficient information regarding COVID-19 [9].

However, the perspectives of market women on COVID-19 in different countries are not well documented. Comparison of the perspectives of market women in different jurisdictions of Africa could be important drivers for governmental policy decisions on the continent. A classic example is the formulation of policies to reduce the negative impact of COVID-19 on the health and business of market women. The choice of the two countries under investigation was due to the structure of the respective economies. Ghana has a relatively informal economy compared to South Africa. Considering the structure of the two economies, it is inherent to access the views of market women in these two countries due to the differences therein. This study therefore aimed to explore the perspectives of market women on COVID-19 in both Ghana and South Africa.

## 2. Materials and Methods

### 2.1. Study Design

A cross-sectional mixed method (embedded approach) was employed for this study. Analysing both qualitative and quantitative data assists the researcher in deeply understanding the research problem which leads to the drawing of salient conclusions [10].

### 2.2. Study Location

The study took place in the new Agogo Market, Ashanti region of Ghana, and Ladysmith market, KwaZulu-Natal Province, South Africa. The new Agogo Market is located in the Asokwa Municipal Assembly about 12 km from Kumasi, the Ashanti regional capital. The Asokwa municipal assembly has Asokwa as its capital and a population of about 125,642 [11]. The Ashanti region is located in the middle belt of Ghana with around 5 million people [12]. The new Agogo Market has about 350 market women. The items sold by the market women include root and tuber crops, grains, legumes, plantain, fruits, vegetables, fresh fish, smoked fish, utensils, clothing, household appliances and building materials. The market is open throughout the week, but Mondays and Fridays are usually busy. The Ladysmith Market is located in Enambithi Municipality about 170 km from the provincial capital (Pietermaritzburg) of the KwaZulu-Natal Province of South Africa [13].

It is a category B municipality within the uThukela District Municipality with a population of about 237,437. Category B refers to the municipality that shares both municipal executive and legislative authority in its area with a category C within whose area it plunges. Category C denotes a municipality that has both municipal executive and legislative authority that consist of more than one municipality [14]. The KwaZulu-Natal (KZN) Province is situated in the southeast part of South Africa with a population of about 11.5 million. The KZN borders the Eastern Cape, Free State and Mpumalanga provinces and Lesotho [13]. The Ladysmith Market has about 200 market women. Vegetables, fruits, mealies and clothes are some of the items sold by the market women in Ladysmith. The market is open throughout the week but gets busy on Fridays and Saturdays.

### 2.3. Participant Selection

Market women aged 18 years and above in both Ghana (new Agogo Community Market) and South Africa (Ladysmith Market) were random/purposively approached at their places of work to participate in the face-to-face interviews using researcher-administered questionnaires. Market women below the age of 18 years as well as those exhibiting signs and symptoms related to COVID-19 during the face-to-face interviews were excluded.

### 2.4. Sample Size Calculation

The population of market women at the new Agogo Community Market (Ghana) and Ladysmith (South Africa) Market are about 350 and 200, respectively. Assuming an error of 5%, a confidence level of 95% and a response distribution of 50%, the sample sizes for both markets were 198 and 130, respectively [15]. The 50% response distribution was considered based on the pilot study that was carried out.

### 2.5. Sampling Technique

Market women were sampled using systematic random sampling [16] by making use of their store numbers and table numbers in both Ghana and South Africa, respectively. The first three of each five stores of the market were sampled for the interview until the desired sample size was reached in the study site (new Agogo Community Market in the Ashanti region of Ghana). In the South African Ladysmith Market, KwaZulu-Natal Province, market women were sampled based on their table numbers as opposed to the store numbers, the technique employed in Ghana. Leaders of the various market women associations in both Ghana and South Africa were approached to guide the team of researchers to the market women in the identified study sites.

### 2.6. Ethical Approval

Ethical approval was obtained from the Committee on Human Research, Publications and Ethics (CHRPE) of the Kwame Nkrumah University of Science and Technology (KNUST) in Kumasi, Ghana (reference number: CHRPE/AP/060/21) and Sefako Makgatho University Research Ethics Committee (SMUREC), South Africa (reference number: SMUREC/P/120/2021:IR).

### 2.7. Data Collection

The study took place in both Ghana and South Africa concurrently between March 2021 and December 2021. Market women in both Ghana and South Africa were visited at their places of work for data collection. All COVID-19 protocols were observed during data collection to minimize the risk of infection between the research assistants and the market women. Face-to-face interviews with the help of semi-structured questionnaires were used to collect data from the research participants after informed consent was obtained. The researchers administered a semi-structured questionnaire developed in an English language but translated into Twi (Ghana) and isiZulu languages (South Africa) languages. The developed semi-structured questionnaire was pilot tested with market women to determine its validity. The pilot study included about 20 market women in both Ghana and South Africa to determine the viability of the intended study to be carried out. The responses received from the market women in both Ghana and South Africa helped us modify the semi-structured questionnaire which was used in this study. The first part of the semi-structured questionnaire collected information on the socio-demographics (age, educational status, years of working experience and average monthly profit). The second part of the questionnaire obtained information on their general understanding of COVID-19, signs and symptoms of the disease, natural therapies employed by them to fight the disease, measures put in place to fight the disease at the workplace and their general perspectives about the impact of COVID-19 on their businesses and the economies of their respective countries (Appendix A).

In Ghana, the face-to-face interviews were conducted by a research assistant who was fluent in both Twi and English languages. In South Africa, the research assistant who conducted the interviews was fluent in both isiZulu and English languages. Each interview lasted for about forty minutes.

### 2.8. Data Analysis

The interviews conducted in both Ghana and South Africa with the market women were conducted in Twi and isiZulu languages, respectively. The interviews were transcribed verbatim. Experts in both Twi and isiZulu were hired to translate the transcripts to the English language. Back translations were conducted by independent language consultants to determine the correctness of the transcripts (from the local languages isiZulu/Twi to English). To ensure the credibility of the study, market women in both Ghana and South Africa were invited by the researchers to confirm the contents of the transcripts [17].

Based on Tesch’s method [18], the texts were then coded and analysed. In summary, the texts were written down. Secondly, an attempt was made to obtain the general perspectives of the market women in this study. Thirdly, similar topics in the analysed text were grouped. Subsequently, the topics were then abbreviated, and the codes were written down. The identified topics from the previous step were ushered into categories. In the next step, a decision was made on each abbreviated category and codes were alphabetized. Preliminary analysis was then carried out to arrange the data belonging to each category during the final stage of the data analysis [18]. Quantitative data analysis was performed using the IBM SPSS version 27.0 (Armonk, NY, USA) [19]. Numerical values (age) and years of experience were expressed as mean ± standard deviation, while other parameters were expressed in frequencies (n) and percentages (%).

## 3. Results

### 3.1. Demographics of Market Women

Table 1 exhibits the demographics of market women from both Ghana and South Africa.

The average age of market women in Ghana was 39.30 ± 11.76 while in SA it was 43.72 ± 9.24. The majority of market women in both Ghana (n = 61, 30.7) and South Africa (n = 81, 90.0%) had completed secondary education. The mean age of years of working experience stood at 9.22 ± 7.38 and 9.93 ± 6.31 for Ghana and South Africa, respectively. The average monthly profit recorded for market women in Ghana was 590.37 ± 752.38 Ghana Cedis which is approximately 90 USD, while South African market women registered an average monthly profit of 5127.30 ± 2297.94 Rand approximately 338 USD. Averagely, Ghanaian market women were making a profit of 3 USD which is higher than the daily minimum daily wage (2.07) announced by the Government of Ghana in 2021 [20]. In South Africa, market women were making a daily profit of about 11 USD which is higher than the daily minimum wage of 1.42 USD announced by the Government of South Africa in 2022 [21].

### 3.2. Understanding the COVID-19 Pandemic

The women expressed their understanding of what COVID-19 was, their statements varied from it being a “global pandemic” (44.72%) to a killer disease (0.50%) in Ghana (GH) and “deadly flu affecting lungs” (67.77%) in South Africa (SA), as summarized in Table 2.

The majority of the market and women in Ghana described it as a global pandemic (n = 89, 44.72%). Some described it as an airborne disease (n = 38, 19.09%) with a few describing it as a viral disease (n = 7, 3.51%). An overwhelming majority of market women in South Africa described COVID-19 as a deadly flu (n = 61, 67.77%).

Table 2 showcases Ghanaian and South African market women’s understanding of COVID-19.

The following statements were made by market women in both Ghana and South Africa regarding their understanding of COVID-19.

“…COVID-19 is a global pandemic.”GH 5.

“This is a foreign disease, it originated from China”GH17.

“COVID-19 is a viral disease.”GH 48.

“It is an airborne disease.”GH 30.

“Deadly flu affecting the lungs.”SA 10.

“COVID-19 is a pandemic flu.”SA 81.

### 3.3. Perceived Signs and Symptoms of COVID-19

The perceived signs and signs of symptoms of COVID-19 mentioned by the market women are exhibited in Table 3.

“To me, cough and fever are the signs that you have COVID-19.”GH 4.

“Catarrh and headache are some of the signs and symptoms of COVID-19.”GH 59.

“Some of the signs and symptoms of COVID-19 are cough, fever, sweating and difficulty to breathe.”SA 8.

“Cough and body pains are the symptoms associated with COVID-19.”SA 30.

### 3.4. The Perceived Mode of Transmission of COVID-19 within Our Communities

Table 4 exhibits the interviewees’ perceived mode of transmission of COVID-19 within our communities. An overwhelming majority of market women in both Ghana (n = 196, 98.49%) and South Africa (n = 90, 100.00%) were able to mention at least one of the following (coming into contact with an infected person, not wearing masks and refusal to sanitize hands) as some of the possible ways in which COVID-19 is transmitted within the communities.

“Not wearing masks and not washing of hands”GH 17.

“Not wearing nose masks and not washing of hands”GH 36.

“Not wearing masks, not washing hands and not practising social distancing”SA 2.

“Not wearing nose masks and not practising social distancing”SA 14.

### 3.5. Natural Therapies Employed by Market Women to Manage COVID-19

The natural therapies employed by the participants to mitigate the effects of COVID-19 are represented in Table 5.

Most of the market women in Ghana (n = 184, 92.46%) mentioned at least one of these foodstuffs (lemon, garlic, ginger and spices) and medicinal plants such as *Azadirachta indica* (neem) tree and *Hibiscus sabdariffa* L. “sobolo” as those they rely on to fight COVID-19. Similarly, an overwhelming majority of market women in South Africa (n = 90, 100.00%) mentioned these foodstuffs (lemon, garlic and ginger) and herbs such as *Artemisia afra* as those they lean on to fight the pandemic. In addition, market women in South Africa emphasized the need to drink hot water regularly to help fight COVID-19.

According to the market women, the therapies and foodstuffs employed by them to fight COVID-19 were as follows:

“I used to drink “sobolo” tea and boiled neem tree leaves to fight the virus.”GH 182.

“I leaned on these foodstuffs (garlic and ginger) to boost my immune system.”GH 82.

“These foodstuffs (lemon and herbs) I ate every day help me to boost my immune system.”SA 87.

“I depend on lemon, garlic, ginger and hot water to fight COVID-19.”SA 82.

### 3.6. Strategies to Fight COVID-19 at the Workplace

Table 6 represents the strategies put in place by the market women to protect themselves and their customers at their places of work.

To access their basic knowledge about the measures put in place at their workplace to limit the spread of the virus, market women in both Ghana and South Africa stated that they rely on the safety protocols such as the wearing of nose masks, hand washing, social distancing and the use of sanitisers.

### 3.7. The General View about the Impact of COVID-19 on Their Businesses

An overwhelming majority of the market women in both Ghana (n = 194, 97.48%) and South Africa (n = 71, 78.88%) indicated a reduction in income as the main consequence of COVID-19. Some market women in South Africa (n = 19, 21.11%) mentioned the collapse of businesses as a consequence of the disease.

The participants’ observations about the impact of COVID-19 on their businesses are represented in Figure 1.

### 3.8. General Knowledge about the Impact of the COVID-19 Pandemic on the Country’s Economy

The participants’ general views about the impact of COVID-19 on their respective country’s economies are represented in Figure 2. Market women in both Ghana (n = 190, 95.47%) and South Africa believed COVID-19 (n = 62, 68.89%) contributed significantly to the negative economic outlook in both countries.

### 3.9. Recommendations to Improve the Profit Margins of Businesses

Table 7 indicates market women’s suggested ideas about the approach to be undertaken by the two respective governments to assist in the improvement of their profit margins during the COVID-19 pandemic. Market women in both Ghana (n = 111, 55.77%) and South Africa (n = 79, 87.77%) suggested the need for their respective governments to provide them with financial assistance to help cushion their loss of revenue due to the pandemic. Market women in Ghana (n = 84, 42.21%) indicated that the increase in fuel prices was a threat to their businesses.

Market women thought that the respective governments must reduce taxes and assist them with loans to support their businesses.

“Our government must help us by reducing taxes”GH 198.

“Loans must be readily accessible to cushion our businesses”GH 181.

“The government must provide loans to support our businesses”SA 29.

“The government must reduce taxes……”SA 9.

## 4. Discussion

Our data analysis has shown that the market women in both Ghana and South Africa have a basic understanding about the COVID-19 pandemic and its negative impact on their businesses and the economy of their respective countries. This is evident in the responses they gave. Regarding the participants’ general understanding of COVID-19, market women’s descriptions of COVID-19 in both Ghana and SA is congruent with scientific literature which described the disease as a viral disease and associated with flu-like symptoms [22].

Market women in both Ghana and South Africa had a basic knowledge of the perceived signs and symptoms of COVID-19. An overwhelming majority of market women in Ghana (n = 186, 93.47%) and South Africa (n = 90, 100.00%) mentioned that a cough, fever, body pains, catarrh, sore throat, cold, sweating difficulty breathing were signs and symptoms of the disease. The aforementioned perceived signs and symptoms of COVID-19 by the participants are in agreement with scientific literature [23]. This may be attributed to the educational public health measures undertaken by both countries during the initial outbreak of the pandemic

The majority of market women in both Ghana and SA mentioned the same foodstuff and therapies, which they relied upon to fight COVID-19. Interestingly, Nie et al., 2021, found that *Artemisia afra* has been proven to be effective against SARS-CoV-2 and feline coronavirus (FCoV) [24], Similarly, Takeda et al., 2020, found that *Hibiscus sabdariffa* L.“sobolo” employed by market women in Ghana to manage COVID-19 has been proven scientifically to possess inhibitory activity against human A virus (IAV) [25]. Workers reported the antiviral activities of the *Azadirachta indica* (neem) tree. The study found that seeds and leaves of *Azadirachta indica* (neem) showed antiviral activity against herpes simplex virus type 1 [26]. *Allium sativum* (garlic) has been proven to be effective against the influenza virus [27]. A study revealed that fresh ginger (*Zingiber officinale*) was effective against the human respiratory syncytial virus in human respiratory tracts [28]. The recommendation by South African women regarding the use of warm water to fight COVID-19 has been emphasized by scientific findings in a previous study carried out by workers [29]. A study conducted by Ramirez et al., 2021, found that hydrotherapy could assist in the prevention and treatment of COVID-19. The same authors suggested that the application of warm water both internally and externally in homes will help people fight the surge of the virus [29].

The market women in both Ghana and South Africa were able to mention the basic protocols (wearing of masks, handwashing, sanitising and social distancing) that must be observed to fight the COVID-19 pandemic. This interesting observation seems to suggest that the respective governments’ health and communication teams’ message about what must be carried out (protocols) to fight the pandemic has reached the grassroots in both countries. This is a major boost towards the two respective countries’ quest to become COVID-19-free nations in the distant future.

The market women in both Ghana and South Africa bemoaned the negative impact of COVID-19 on their business due to the loss of income as a result of the lockdown restrictions imposed by the respective governments in these two countries. This is in agreement with a study conducted in Zimbabwe where women entrepreneurs had no option but to close their businesses due to the loss of revenues as a result of COVID-19 lockdown restrictions imposed by the Zimbabwean government [30]. Similarly, in Bangladesh due to the hard lockdown policy implemented by the government to mitigate the effects of COVID-19, about 7 million Bangladeshi citizens working in small–medium-sized enterprises lost their jobs [31]. In Russia, Kuvalin et al., 2021 [32], found that a large proportion of enterprises (more than 50%) experienced a huge decline in profit due to COVID-19. In South Africa, informal traders’ inabilities to obtain a permit to sell their goods at the market due to the South African government’s stricter measures to contain COVID-19 harmed their businesses [33]. A study carried out in Ghana revealed that hard lockdown restrictions had a serious negative impact on the informal traders’ fragile businesses, which plunged most of them into extreme poverty [34].

Market women in both Ghana and South Africa spoke about the negative impact of COVID-19 on their respective country’s economies. Available data suggest that Ghana had a decrease in economic growth from 5.8% to 1.5% [35], while the GDP growth of South Africa dwindled by about 5% by the year-end of 2020 [36]. Market women in Ghana spoke about the increasing prices of fuel in their country which have had a serious negative impact on their businesses. They pleaded with the current government in Ghana to make sure that fuel prices are drastically reduced. There was a huge public outcry about the economic hardships experienced by the Ghanaian population due to the increase in fuel prices. For example, a general strike by commercial drivers to protest the fuel hikes left commuters stranded for a day which had a serious negative impact on the nation’s economy [37]. The market women in South Africa (n = 15, 7.89%) specifically talked about the increasing prices of food items in their country. Agyei et al., 2021, found that there was an increase in food prices in most of the countries within sub-Saharan Africa due to COVID-19 [38]. The interviewees in both Ghana and South Africa asked for governmental financial assistance to mitigate the effect of the pandemic on their businesses, such as the financial assistance implemented in Nepal and Tajikistan. In Nepal, food prices are fixed to protect vulnerable women food producers [39]. Similarly, in Tajikistan, the local government provides soft loans to women entrepreneurs in the food and medical supplies sectors [40].

### 4.1. Recommendations for Further Studies and Interventions

Studies have shown that compared to men, women entrepreneurs are generally affected during a pandemic because they are more susceptible to difficult situations [41,42]. The findings of this study will assist the various respective governments on the African continent to implement policies that will limit the economic hardships women faced during a pandemic. The authors of this study proposed the following interventional policies to be adopted by the various governments on the African continent. Firstly, there should be an establishment of the Women Anti-pandemic Support Fund (WASF). This consolidated fund (WASF) should be managed by an established secretariat. A proportion of the budget of each of the two countries (Ghana and South Africa) should be allocated for the running of the consolidated fund which will be used to support women during a pandemic. Secondly, the governments of both Ghana and South Africa should provide tax-free soft loans to women affected during the COVID-19 pandemic. Thirdly, the department of trade and industry in conjunction with the department of employment and social welfare in both countries should conduct workshops and seminars to educate market women about backyard farming, network marketing, dressmaking, etc., which will allow them to earn extra income. Lastly, the respective governments in both Ghana and South Africa must establish a fuel pricing committee under the department of energy and natural resources to ensure that consumers pay realistic prices for fuel products all year round.

### 4.2. Limitations of This Study

This study was carried out in two countries, Ghana and South Africa. To understand the problem from a wider perspective, the study should be replicated in other countries. The fear of being infected with COVID-19 by getting into contact with different people among the studied population should have contributed to the low sample size recorded in South Africa. The study did not consider vaccination status of the participants, which would have added to the body of knowledge on vaccine uptake in this population.

## 5. Conclusions

This study explored market women’s perspectives about COVID-19 on the African continent. The data suggest that market women in both Ghana and South Africa had a good general understanding of COVID-19 and its effects on their livelihoods and their respective country’s economies. Market women’s description of COVID-19 as a disease associated with flu-like symptoms is consistent with scientific literature. Market women’s perception of the mode of transmission of COVID-19 (being in contact with an infected person) is congruent with the scientific literature. Natural remedies such as lemon, garlic and ginger employed by the market women to mitigate the effects of COVID-19 have been scientifically proven to possess antiviral activities. The strategies (wearing of masks and handwashing) adopted by market women to limit the transmission of the virus at their workplaces conform to the safety protocols adopted by the World Health Organization (WHO) to fight the pandemic. The market women believed that COVID-19 had harmed their businesses and the economies of their respective countries. Market women in both Ghana and South Africa pleaded with their respective governments to make loans readily accessible to cushion their businesses.

## Figures and Tables

**Figure 1 ijerph-19-09658-f001:**
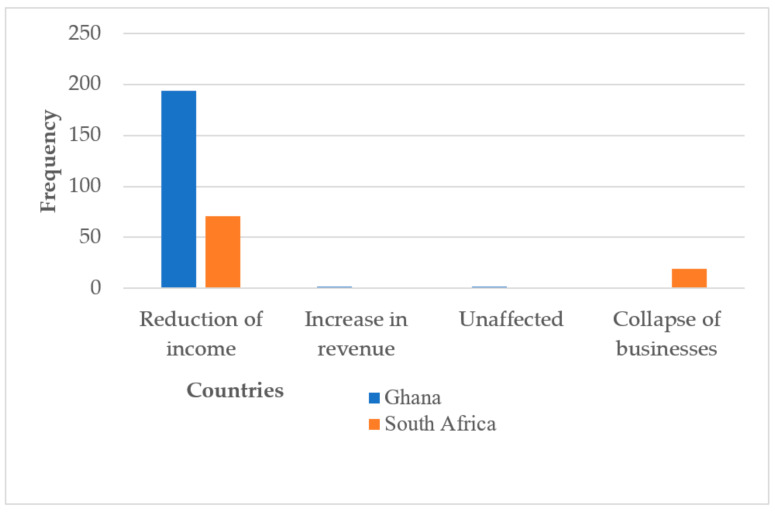
Market women’s general view about the impact of COVID-19 on their businesses.

**Figure 2 ijerph-19-09658-f002:**
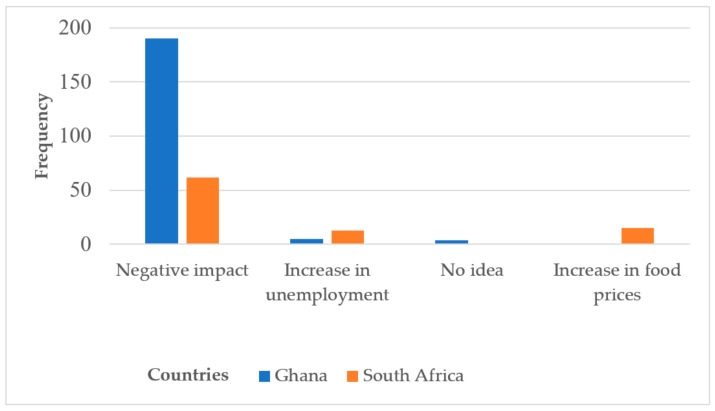
Market women’s opinions about the impact of COVID-19 on their countries.

**Table 1 ijerph-19-09658-t001:** Socio-demographic characteristics of Ghanaian and South African market women.

Variables	(Ghana *N* = 199) No.%	(South Africa *N* = 90) No.%	
Age (Years)			
18–28	35 (17.60)	2 (2.20)	
29–39	56 (28.10%)	30 (33.30%)	
40–50	62 (31.20%)	39 (43.30%)	
51–61	38 (19.10%)	17 (18.90%)	
62–72	8 (4.00%)	2 (2.20%)	
Mean (SD)	39.30 ± 11.76	43.72 ± 9.24	
Education level			
Nil	61 (30.70%)	0 (0.00%)	
P.E	58 (29.10%)	5 (5.50%)	
S.E	61 (30.7%)	81 (90%)	
T.E	19 (9.50%)	4 (4.50%)	
Years of work experience			
1–10	132 (66.3%)	67 (74.40%)	
11–20	56 (28.1%)	17 (18.90%)	
21–30	9 (4.5%)	5 (5.6%)	
31–40	2 (1.0%)	1 (1.1%)	
Mean (SD)	9.22 ± 7.38	9.93 ± 6.31	
Monthly business profit (GHC)	(Ghana *N* = 199) No.%	ZAR	(South Africa *N* = 90) No.%
0–1000	179 (89.95%)	1000–5000	53 (58.90%)
1001–2000	14 (7.00%)	5001–10,000	35 (38.90%)
2001–3000	2 (1.00%)	10,001–15,000	2 (2.20%)
3001–4000	2 (1.00%)	N/A *	N/A *
4001–5000	2 (1.00%)	N/A *	N/A *
Mean (SD)	590.37 ± 752.38	Mean (SD)	5127.30 ± 2297.94

Legend: Primary education (P.E); Secondary education (S.E); Tertiary education (T.E); Ghana cedis (GHC); South African Rand (ZAR); Nil (Had no formal education); N/A * (Not applicable).

**Table 2 ijerph-19-09658-t002:** Market women’s understanding of COVID-19.

Variable	Ghana (*N* = 199) No. (%)	South Africa (*N* = 90) No. (%)
Understanding of COVID-19		
Global pandemic	89 (44.72%)	N/A *
Deadly flu affecting lungs	N/A *	61 (67.77%)
Airborne disease	38 (19.09%)	
Pandemic flu	N/A *	29 (14.57%)
Foreign disease	34 (17.08%)	N/A *
Deadly disease	16 (8.04%)	N/A *
Communicable disease	9 (4.52%)	N/A *
Viral disease	7 (3.51%)	N/A *
Not real	3 (1.50%)	N/A *
No idea	2 (1.00%)	N/A *
Killer disease	(0.50%)	N/A *

N/A *—not applicable.

**Table 3 ijerph-19-09658-t003:** Perceived signs and symptoms of COVID-19.

Variable	Ghana (*N* = 199) No. (%)	South Africa (*N* = 90) No. (%)
Perceived signs and symptoms of COVID-19		
Cough, fever, headache, Catarrh, sore throat, cold, sneezing	188 (93.47%)	N/A *
Cough, fever, cold, sore throat, dizziness, catarrh, headache pains, difficulty breathing and sweating	N/A *	90 (100%)
Anosmia	8 (4.02%)	N/A *
Bad breath	1 (0.50%)	N/A *
Loss of sense of smell	1 (0.50%)	N/A *
Dyspnoea	1 (0.50%)	N/A *

N/A *—not applicable.

**Table 4 ijerph-19-09658-t004:** Perceived mode of transmission of COVID-19 within our communities.

Variable	Ghana (*N* = 199) No. (%)	South Africa (n = 90) No. (%)
Mode of transmission of COVID-19 in our communities		
Airborne and physical contact with an infected person	194 (97.48%)	N/A *
Physical contact, not wearing masks, not sanitising, not practising social distancing and not washing hands	N/A *	90 (100.00%)
Not practising social distancing	2 (1.00%)	N/A *
No idea	2 (1.00%)	N/A *
Not real	1 (0.50%)	N/A *

N/A *—not applicable.

**Table 5 ijerph-19-09658-t005:** Foodstuffs and therapies employed by market women to manage COVID-19.

Variable	Ghana (*N* = 199) No. (%)	South Africa (*N* = 90) No. (%)
Therapies used to manage COVID-19		
The neem tree, *Hibiscus sabdariffa* L. “sobolo” vegetables, ginger, garlic, fruits, lemon, local spices, moringa	184 (92.46%)	N/A *
Lemon, garlic, ginger, herbs, Artemisia, warm water	N/A *	90 (100.00%)
Herbal medicine	6 (3.01%)	N/A *
Hot food	4 (2.01%)	N/A *
No idea	5 (9.95%)	N/A *
Known benefits of the identified foodstuff to manage COVID-19		
Immune booster	179 (89.95%)	52 (57.77%)
Kills virus	9 (4.52%)	38 (42.22%)
Prophylactic	3 (1.50%)	N/A *
No idea	8 (4.02%)	N/A *

N/A *—not applicable.

**Table 6 ijerph-19-09658-t006:** Strategies put in place to fight COVID-19 at the workplace.

Variable	Ghana (*N* = 199) No. (%)	South Africa (*N* = 90) No. (%)
Protocols observed at the workplace to mitigate the effect of COVID-19		
Wearing masks, hand washing, sanitisers	N/A *	53 (58.80%)
Sanitisers, wearing of masks, social distancing	N/A *	23 (25.50%)
Sanitisers, wearing masks	30 (15.07%)	14 (15.50%)
Wearing masks, hand washing	47 (23.62%)	N/A *
Hand washing, sanitisers	14 (7.03%)	N/A *
Wearing of masks, social distancing	13 (6.53%)	N/A *
Hand washing	6 (3.01%)	N/A *
Wearing of masks	18 (9.05%)	N/A *
Sanitisers	10 (5.03%)	N/A *
Social distancing	8 (4.02%)	N/A *

N/A *—not applicable.

**Table 7 ijerph-19-09658-t007:** Recommendations to improve the profit margins of businesses.

Variable	Ghana (*N* = 199) No. (%)	South Africa (*N* = 90) No. (%)
Recommendations to improve the profit of market women businesses		
Provision of loans by the government	111 (55.77%)	79 (87.77%)
Reduction in fuel prices and taxes	84 (42.21%)	N/A *
Reduction in taxes	N/A	9 (10.00%)
Lifting of lockdown restrictions	3 (1.50%)	
No idea	N/A	2 (2.22%)

N/A *—not applicable.

## Data Availability

Not applicable.

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
