# Peer review of "Market Women’s Perspectives on Coronavirus Disease 2019 (COVID-19): The Case of Ghana and South Africa"

_ijerph, 2022, doi:10.3390/ijerph19159658_

Round 1
Reviewer 1 Report
The authors have made a very complete revision, incorporating all the contributions made in a very appropriate manner.
I attach in the cover letter sent by the authors with the corrections, small contributions, especially in terms of format, for the authors to take into account. I have highlighted them in green.
I still consider that the fact of quantifying data (grouping in frequencies and percentages) the information that has been collected through open questions is a controversial issue that raises doubts about the consistency of the methodology. In any case, if the editor of the journal considers that this is not such a relevant issue, I do not have much more to say about it.
Kind regards

Author Response
Dear Editor please find attached the authors' response to reviewer 1

Reviewer 2 Report
Thank you for the opportunity to review the article entitled Market Women’s Perspectives on Coronavirus disease 2019 2 (COVID-19): The Case of Ghana and South Africa, which addresses the main causes of COVID-19 vaccination hesitancy and its particularities across the different regions of South Africa abnd West Africa. The results of this study are important because they strengthens previous findings regarding the cluster of factors associated with the knowledge, believes and attitudes associated with COVID-19 pandemic. This understanding may be important to reinforce thepreventive measures against this infection. Transcultural studies are very important because they ofer the possibility to compare different culturally shaped attitudes and behaviors which an be adreesed by identifing common determinants of preventive measures and vaccine acceptance.
The study is correctly designed and technically sound. The methods used in this research are well described and provide sufficient details to be understand. The research methodology is in line with the proposed objectives. The results are appropriately interpreted and respond to the hypothesis of the study.
The disscutions address the findings of the research in relation with other studies and also propose potential explanation for different acceptance of general profilacic measures. Also, it will be helpfull to answer to the following quetions:
1. Did you colect data regarding vaccination againest COVID in your study? If so, it will be a befeficial addition to the currently presented results.
2. What are the vaccination rates of the market women ?
3. Are there any correlation between vaccination reate and the market women's general view about the impact of COVID-19 on their businesses?
Thank you for your esteemed efforts in increasing our collective knowledge about COVID-19 preventive measures hesitancy determinants.
Sincerly yours
Author Response
Dear Editor
Please find attached the authors' response to reviewer 2

Reviewer 3 Report
The paper addresses an issue of limited relevance, not excluding that it is very important for the local community. The article does not meet the requirements for a scientific text. It is in fact a popular science text.
Introduction. Very poorly developed. Poorly articulated research problem. Lack of research questions and hypotheses.
Materials and methods: Very small research sample, not representative.
Research instrument: not standardised.
Statistical analysis: basic descriptive statistics. Mainly percentage analysis.
Discussion: adequate to the topic presented.
The issues from the social point of view of African countries may be important, but first the question of what is the research problem must be answered. Then, in accordance with the methodological requirements of scientific research, select the explanatory (independent) variables, set hypotheses and operationalise them.
The text must therefore be written from scratch.
Author Response
Dear Editor
Please find attached the authors' response to reviewer 3

Round 2
Reviewer 2 Report
Dear authors,
thank you very much for your honest response. I understand that you didn't colect data regarding vaccination rate in the participants of the study and I would recommend to mention this as a limitation of your study.
Sincerly yours,
Author Response
Please find attached the author's response to reviewer 2

This manuscript is a resubmission of an earlier submission. The following is a list of the peer review reports and author responses from that submission.
Round 1
Reviewer 1 Report
The restrictions on the people's global movement, commodities, and services, and the measures taken to reduce COVID-19 spread, have disrupted food environments around the world and forced us to collectively redesign and optimize our systems using the existing resources in a more sustainable perspective. This paper aimed to explore the perspectives of market women on COVID-in both Ghana and South Africa.
The manuscript addresses an interesting topic and prior to publication only some minor English corrections are required. The topic is not very complex, but I think it is interesting, so I recommend publishing it, after correcting spelling mistakes.
Reviewer 2 Report
Abstract
The abstract needs to be written much more concisely, while highlighting the newly discovered points.
- Materials and methods
Two countries were selected as the study area. Is there a specific reason for choosing these areas? It seems that it will arouse the reader's interest more if describing the clear answers to this question than to simply describe the region.
- Results
Table 1 seems to need correction as some parts cannot be observed. When discussing average income in your study sample, it is necessary to describe how much it compares to the average income of each country.
Table 2 compares the two countries, and it appears that there may have been some problems in the interviews when collecting data. It is not very convincing that South Africans answered with only two items when the same questionnaires were presented to both countries. The explanation related to the table need to go beyond a simple description and requires a detailed discussion of this part.
Although this study analyzes two countries, each country is simply interpreted separately when interpreting the results. It seems necessary to interpret the results in consideration of differences and similarities in each country's system or environment.
Reviewer 3 Report
This is an interesting study that assesses the perspective of market women on COVID in two different countries, Ghana and South Africa.
The article reflects the effort made to carry out the project under very difficult conditions, due to the pandemic situation in which the fieldwork was carried out. For this I would like to express my special recognition.
Despite the effort made, the article has important methodological limitations, including the fact that it attempts to quantify results that have been obtained qualitatively.
In addition to this problem, it has other aspects that should be improved and which I have pointed out in the contributions that I have included, in the form of comments, in the attached document.
From my point of view, the important methodological limitations of the study condition the validity of the results and I consider that the article should not be accepted.
Kind regards
